# STADE: STANDARD DEVIATION AS A PRUNING METRIC

## ABSTRACT

Large Language Models (LLMs) have become very widespread and are used to solve a wide variety of tasks. To successfully handle many of these tasks, LLMs require longer training times and larger model sizes. This makes LLMs ideal candidates for pruning methods that reduce computational demands while maintaining performance. Previous methods require a retraining phase after pruning to maintain the original model's performance. However, state-of-the-art pruning methods, such as Wanda, prune the model without retraining, making the pruning process faster and more efficient. Building upon Wanda's work, this study provides a theoretical explanation of why the method is effective and leverages these insights to enhance the pruning process. Specifically, a theoretical analysis of the pruning problem reveals a common scenario in Machine Learning where Wanda is the optimal pruning method. Furthermore, this analysis reveals cases where Wanda is no longer optimal. To tackle those cases, we develop a new method, *STADE*, based on the standard deviation of the input. From a theoretical and empirical standpoint, *STADE* demonstrates better generality across different scenarios. Finally, extensive experiments on Qwen, Llama and Open Pre-trained Transformers (OPT) models validate these theoretical findings, showing that depending on the training conditions, Wanda's optimal performance varies as predicted by the theoretical framework.

## 1 INTRODUCTION

Large Language Models (LLMs) (Radford et al., 2018; 2019; Brown et al., 2020) have revolutionized not only the field of Natural Language Processing (NLP) but also numerous real-world applications that affect everyday life. Their ability to generate coherent text, perform complex reasoning, and support a variety of conversational and decision-making tasks has led to widespread adoption in both research and industry. With the advent of increasingly autonomous systems (Durante et al., 2024; junyou li et al., 2024; Wu et al., 2024), these models now assist with tasks ranging from content creation and translation to automated customer support and strategic decision making.

Despite these impressive capabilities, LLMs are notorious for their substantial computational requirements (Kaplan et al., 2020a). The high memory footprint, extensive processing power, and significant energy consumption often limits their deployment on devices with limited resources, such as mobile phones or embedded edge devices. In addition, the large-scale training of these models contributes to increased operational costs and a non-negligible environmental impact. Consequently, the drive to reduce the computational and storage demands of LLMs has become a central focus in the field (Sevilla et al., 2022).

To mitigate these computational challenges, a variety of approaches have been explored. One prominent strategy involves reducing the storage requirements of model weights through *quantization* (Ma et al., 2024; Wu et al., 2020). Quantization techniques lower the numerical precision of weights and activations, resulting in reduced memory usage and accelerated inference speeds, often with minimal degradation in performance. Another effective approach is to remove unimportant weight parameters through *pruning* (LeCun et al., 1989). Pruning methods seek to eliminate redundancies in the network by removing weights that contribute little to overall model performance, thereby decreasing both the computational load and the inference latency.

Pruning techniques can be applied during training (Sanh et al., 2020) or after the model has been fully trained, in what is known as *post-training pruning* (Ashkboos et al., 2024). The latter approach is particularly appealing when the goal is to adapt a pre-trained model for deployment on resource-constrained devices, as the main challenge is not the training process but rather fitting the model into a limited hardware environment. Although some post-training pruning strategies involve costly retraining steps (Agarwal et al., 2024; Xu et al., 2024), previous studies (Sun et al., 2024; Frantar & Alistarh, 2023) have demonstrated that a model can maintain a large fraction of its original performance even when 50% of its weights or more are pruned without any retraining.

A notable pruning method is Wanda (Sun et al., 2024), which employs a simple yet effective strategy based on the $L_2$-norm to guide weight removal. Despite its empirical success, the fundamental reason for the superior performance of the $L_2$-norm over alternative norms (e.g., $L_1$ or $L_\infty$) was neither formally analyzed or fully understood. As noted in the original paper: *"We find that $L_2$-norm tends to work better than other norm functions (e.g., $L_1$ and $L_\infty$) in measuring activation magnitudes. This is possibly because $L_2$-norm is generally a smoother metric"* (Sun et al., 2024). Such observations have motivated deeper theoretical investigations into pruning criteria.

This work aims to provide a comprehensive analysis of the pruning problem. The contributions are as follows:

- A theoretical analysis of the pruning problem is presented, revealing a characterization of machine learning scenarios where *Wanda* emerges as the optimal pruning method.
- The analysis is extended to cases where *Wanda*'s approach is suboptimal, thereby motivating the development of a new method, *STADE*.
- Multiple experiments with different LLM model families validate empirically the theoretical analysis.
- Additionally, an ablation of layer-specific characteristics demonstrates that different layers benefit more from using different pruning metrics depending on the input characteristics. To the best of our current knowledge, this is the first work to apply distinct pruning metrics to different layers, resulting in improved overall pruning effectiveness.

Extensive experiments have been performed across multiple models and configurations to validate the theoretical insights and assess the performance of the proposed *STADE* method. The experiments evaluate perplexity and zero-shot capabilities for various models with different pruning metrics, and reveal that the impact of pruning is highly dependent on the statistical properties of the input at each layer.

## 2 RELATED WORK

The study of sparse subnetworks within large neural networks has been an area of intense investigation, particularly following the introduction of the *Lottery Ticket Hypothesis* (Frankle & Carbin, 2019). This hypothesis proposes that within a randomly initialized neural network there exist subnetworks (or "winning tickets") that, when trained in isolation, can achieve performance on par with the full network. Subsequent investigations (Morcos et al., 2019; Frankle et al., 2020) have further elucidated the generalization capabilities and connectivity properties of these subnetworks, providing a theoretical basis for pruning methods.

Pruning strategies have evolved significantly over the past decade. Early methods relied on simple heuristics such as magnitude-based pruning (Zhu & Gupta, 2017), which removes weights with the smallest absolute values under the assumption that these contribute least to network performance. This basic approach laid the groundwork for more sophisticated techniques that consider additional information about the network. For instance, the work in (Han et al., 2015) utilized the $L_2$-norm to evaluate the importance of weights, demonstrating that many redundant parameters could be pruned without significant loss in accuracy.

Advancements in pruning have also led to the development of methods that incorporate second-order information. The Optimal Brain Surgeon (OBS) algorithm (Dong et al., 2017) leverages the Hessian matrix of the loss function to estimate the impact of removing individual weights. Although OBS provides more refined pruning decisions, its high computational complexity has restricted its practical application in large-scale models.

More recent approaches have shifted focus to dynamic pruning strategies that are integrated into the training process (Sanh et al., 2020; Chen et al., 2021). These methods progressively reduce the number of active parameters during training, often resulting in models that are sparser and more computationally efficient. However, such strategies may conflict with the scaling laws observed for LLMs (Kaplan et al., 2020b), where performance improvements are closely tied to increases in model size, computational resources, and data availability. As a consequence, post-training pruning techniques have emerged as a pragmatic solution for adapting large pre-trained models to resource-limited environments.

A wide range of post-training pruning techniques has been proposed in recent years. Some methods, such as LoRA-based pruning (Zhou et al., 2024), incorporate low-rank adaptations to guide the pruning process. However, retraining the pruned model often incurs significant computational overhead. Others, like SparseGPT (Frantar & Alistarh, 2023), use Hessian-based metrics to carefully select which weights to remove, and adjusting the remaining parameters accordingly, thereby preserving critical network functionality. Additionally, strategies that minimize local reconstruction errors within individual blocks (Agarwal et al., 2024; Bai et al., 2024) or layers (Hubara et al., 2021a) of Transformer-based architectures have been investigated, underscoring the notion that different layers may require tailored pruning criteria. Some layer-wise pruning techniques employ structured sparsity, assigning a learned importance weight to each matrix, thereby determining its optimal sparsity level (Li et al., 2024). Others adopt a block-wise grouping strategy, optimizing sets of layers collectively (Xu et al., 2024) to balance sparsity and accuracy.

A central aspect of all pruning methodologies is the selection of an appropriate pruning metric that accurately distinguishes between essential and redundant weights. The metric adopted in Wanda (Sun et al., 2024)—which involves computing the $L_2$-norm of the input and multiplying it by the absolute value of the corresponding weight—has garnered considerable attention for its simplicity and effectiveness. This approach provides a smooth, continuous measure that captures the contribution of each weight to the overall activations. In contrast, more elaborate metrics use second-order derivatives and update the unpruned weights. These changes not only make it more complex but also increases the computational time and memory required.

Overall, the evolution of pruning methods reflects a broader trend in machine learning towards achieving a balance between model efficiency and predictive performance. Early heuristic methods are principled approaches that take into account the underlying statistics and structure of the network. These previous studies serve as a valuable foundation for the enhancements presented in this work, including the development of the *STADE* method, which refines pruning strategies by incorporating the statistical characteristics of layer inputs.

## 3 METHODOLOGY

Consider a data matrix $X \in \mathbb{R}^{N \times M}$ and a weight matrix $\mathbb{W} \in \mathbb{R}^{M \times H}$, where $N$ is the number of instances in the dataset, $M$ represents the number of features and $H$ represents the number of output features. In Wanda (Sun et al., 2024), the pruning of each column $\mathbb{W}_{:,i} \in \mathbb{R}^M$ is performed according to the criterion:

$$\min_j \|X_{:,j}\|_2 |\mathbb{W}_{j,i}| \tag{1}$$

where $\|X_{:,j}\|_2$ is the $L_2\text{-}norm$ of feature $j$ in the dataset. In the following section, the pruning problem is formalized and it is demonstrated that *Wanda* selection criterion is optimal for layers with a centered inputs, i.e., inputs whose expected value in each coordinate is 0. With this insight, a generalization to layers with uncentered inputs is derived, leading to the proposed method *STADE*.

### 3.1 PROBLEM DEFINITION

Let $X \in \mathbb{R}^M$ be a random multivariate variable with $\mu_i = \mathbb{E}[X_i]$ and $\sigma_i^2 = Var[X_i]$, and consider a linear layer with a weight matrix $\mathbb{W} \in \mathbb{R}^{M \times H}$ and a bias term $\mathbb{B} \in \mathbb{R}^H$. The pruning process for the $m$-th column will be focused on the weight vector (denoted by $W = \mathbb{W}_{:,m} \in \mathbb{R}^M$) and the corresponding bias (denoted by $B = \mathbb{B}_m \in \mathbb{R}$). In this setting, the pruning problem aims to find the optimal $W^* \in \mathbb{R}^M$ and $B^* \in \mathbb{R}$ such that:

$$\min_{W^*, B^*} \mathbb{E}\left[\left((B + \sum_{i=1}^{M} X_i W_i) - (B^* + \sum_{i=1}^{M} X_i W_i^*)\right)^2\right] \tag{2}$$
$$\text{s.t. } \forall i \in \{1, ..., M\}\backslash\{j\}, W_i^* = W_i, W_j^* = 0$$

Note that the objective is to select the pruning weight $W_j$ so that the output remains almost unchanged, while only allowing the bias term to be updated.

### 3.2 STADE DERIVATION

Starting from the formulation in Eq. 2, the objective function can be reformulated as follows:

$$\mathbb{E}\left[\left((B + \sum_{i=1}^{M} X_i W_i) - (B^* + \sum_{i=1}^{M} X_i W_i^*)\right)^2\right]$$
$$= \mathbb{E}[((B - B^*) + X_j W_j)^2]$$
$$= \mathbb{E}[(B - B^*)^2 + 2(B - B^*)(X_j W_j) + (X_j W_j)^2 \tag{3}$$
$$= (B - B^*)^2 + 2(B - B^*)\mathbb{E}[X_j W_j] + \mathbb{E}[(X_j W_j)^2]$$
$$= (B - B^*)^2 + 2(B - B^*)\mu_j W_j + (\sigma_j^2 + \mu_j^2)W_j^2$$

To determine the optimal solution of the convex problem (with respect to $B^*$) in Eq. 3, the derivative is computed to obtain the stationary and minimum point:

$$\frac{d}{dB^*}\left[\mathbb{E}\left[\left((B + \sum_{i=1}^{M} X_i W_i) - (B^* + \sum_{i=1}^{M} X_i W_i^*)\right)^2\right]\right]$$
$$= \frac{d}{dB^*}[(B - B^*)^2 + 2(B - B^*)\mu_j W_j + (\sigma_j^2 + \mu_j^2)W_j^2] \tag{4}$$
$$= -2(B - B^*) - 2\mu_j W_j = 0 \Leftrightarrow B^* = \mu_j W_j + B$$

Substituting the optimal bias in Eq. 3 yields the solution for $W^*$:

$$\min_{W^*, B^*} \mathbb{E}\left[\left((B + \sum_{i=1}^{M} X_i W_i) - (B^* + \sum_{i=1}^{M} X_i W_i^*)\right)^2\right]$$
$$= \min_{j, B^*}(B - B^*)^2 + 2(B - B^*)\mu_j W_j + (\sigma_j^2 + \mu_j^2)W_j^2 \tag{5}$$
$$= \min_j (B - (\mu_j W_j + B))^2 + 2(B - (\mu_j W_j + B))\mu_j W_j + (\sigma_j^2 + \mu_j^2)W_j^2$$
$$= \min_j (\mu_j W_j)^2 - 2(\mu_j W_j)\mu_j W_j + (\sigma_j^2 + \mu_j^2)W_j^2 = \min_j \sigma_j^2 W_j^2$$

Since the goal is to find the weight $W_j$ that minimizes $\min_j \sigma_j^2 W_j^2$, it is enough with finding $arg \min_j$ instead of the $\min_j$. Therefore we can simplify the problem as follows:

$$arg \min_j \sigma_j^2 W_j^2 = arg \min_j \sigma_j |W_j|$$
$$\approx arg \min_j \frac{||X_{:,j} - \frac{1}{N}\sum_{n=1}^{N} X_{n,j}||_2}{\sqrt{N-1}} |W_j| \tag{6}$$
$$= arg \min_j ||X_{:,j} - \frac{1}{N}\sum_{n=1}^{N} X_{n,j}||_2 |W_j|$$

Since the our goal is to find the optimal $j$ that minimizes the loss ($arg \min_j$), the factor $\frac{1}{N-1}$ and the squaring operation can be omitted.

### 3.3 WANDA DERIVATION

Many modern Transformers (Touvron et al., 2023a;b; Dubey et al., 2024) employ normalization layers. This simplifies the original problem by enforcing the input $X$ to be normalized ($\mu_i = \mathbb{E}[X_i] = 0$). This addition to the previous derivations (Eqs. 4 and 5) leads to:

$$B^* = \mu_j W_j + B = 0 * W_j + B = B$$

$$\min_j ||X_{:,j} - \mu_j||_2 |W_j| = \min_j ||X_{:,j}||_2 |W_j| \tag{7}$$

This derivation results in the *Wanda* criterion, where the bias term doesn't need to get updated. Please notice that *Wanda* is optimal under the previous assumptions, i.e., it is only optimal for layers with centered inputs.

### 3.4 STADE-W: USING DIFFERENT METRICS FOR DIFFERENT LAYERS

Based on the previous theoretical insight we introduce *STADE-W*, a pruning strategy that employs different pruning criterions depending on whether the input is normalized. The pruning metrics derived from the previous analysis are as follows:

$$\text{Wanda criterion:} \quad ||X_{:,j}||_2 |W_{i,j}| \tag{8}$$

$$\text{STADE criterion:} \quad ||X_{:,j} - \frac{1}{N}\sum_{n=1}^{N} X_{n,j}||_2 |W_{i,j}| \tag{9}$$

*STADE-W* applies the *STADE* metric for biased inputs (such as the second layer of an MLP or the output layer in multi-head attention) and the *Wanda* metric for unbiased inputs (such as the first layer of an MLP or the queries, keys, and values in multi-head attention). In theory, *STADE* should be able to identify that the mean is 0 and return the same output as *Wanda*. However, in practice the dataset used for calibration might lead to a slightly different mean estimation and therefore, *STADE* ends up underperforming.

Table 1: Comparison of pruning weight metrics across different methods. The column *Centered Input* indicates whether the pruning method distinguishes between inputs with zero mean (Yes), without zero mean (No), or treats them equivalently (Any).

| Method | Weight Update | Centered Input | Pruning criterion |
|---|---|---|---|
| Magnitude (Zhu & Gupta, 2017) | ✗ | Any | $|W_{i,j}|$ |
| Wanda (Sun et al., 2024) | ✗ | Any | $||X_{:,j}||_2 |W_{i,j}|$ |
| Sparsegpt (Frantar & Alistarh, 2023) | ✓ | Any | $[|W|^2/diag[(X^T X + \lambda\mathbf{1})^{-1}]]_{i,j}$ |
| STADE | ✗ | Any | $||X_{:,j} - \frac{1}{N}\sum_{n=1}^{N} X_{n,j}||_2 |W_{i,j}|$ |
| STADE-W | ✗ | Yes | $||X_{:,j}||_2 |W_{i,j}|$ |
| | ✗ | No | $||X_{:,j} - \frac{1}{N}\sum_{n=1}^{N} X_{n,j}||_2 |W_{i,j}|$ |

## 3.5 OPTIMAL PRUNING METRIC

In order to clarify which pruning metric to use in which linear layer we make the following distinctions:

- **Centered inputs**: When the input distribution is centered the optimal method is *Wanda*. This can only be assured when the previous layer is a normalization layer such as *Batchnorm* (Ioffe & Szegedy, 2015), *Groupnorm* (Wu & He, 2018) or *Layernorm* (Ba, 2016), but this is not certain for the *RMSnorm* layer (Zhang & Sennrich, 2019).
- **Uncentered inputs**: In this case, the input mean is no longer 0 and therefore *Wanda* is no longer optimal. Therefore *STADE* should be use since it takes into account the non-zero mean.

The experiments will follow this set-up unless specified otherwise. Notice that within the same model, different layers could belong to different scenarios as mentioned before with *STADE-W*.

## 4 EXPERIMENTS

**Models and Evaluation.** Most experiments are conducted using the Llama (Touvron et al., 2023a;b; Dubey et al., 2024) and Qwen (Bai et al., 2023; Yang et al., 2024; Qwen et al., 2025; Yang et al., 2025) models. In addition, the OPT family (Zhang et al., 2023) is also evaluated due to its architectural differences such as the usage of a bias term in the linear layers, the usage of *Layernorm* (Ba, 2016) and the incorporation of positional embeddings instead of rotary position embeddings (Su et al., 2024).

Following previous research (Sun et al., 2024), C4 dataset (Raffel et al., 2019) is used for calibration, while raw-WikiText2 dataset (Merity et al., 2022) is employed to evaluate model perplexity. Moreover, the zero-shot capabilities of the pruning methods are assessed with eight tasks using EleutherAI LM Harness (Gao et al., 2024) package. These tasks include: *Boolq* (Clark et al., 2019), a yes/no question answering dataset containing 15,942 examples; the *Recognizing Textual Entailment (RTE)* suite, which combines RTE-1 (Dagan et al., 2006), RTE-2 (Dagan et al., 2005), RTE-3 (Delmonte et al., 2007), and RTE-5 (Bentivogli et al., 2009) challenges constructed from news and Wikipedia text; *HellaSwag* (Zellers et al., 2019), a challenging dataset for evaluating commonsense,; *WinoGrande* (Keisuke et al., 2019), a binary fill-in-the-blank task that requires commonsense reasoning; *Arc-Easy* and *Arc-Challenge* (Clark et al., 2018), which consist of multiple-choice science questions targeting grade-school level content and are split into easy and challenging subsets, *OpenBookQA* (Mihaylov et al., 2018), a dataset that involves questions requiring multi-step reasoning, additional commonsense knowledge, and comprehensive text comprehension; and *MMLU* (Hendrycks et al., 2021), a multitask test consisting of multiple-choice questions from various branches of knowledge.

**Baselines.** The main experiments employ pruning methods that do not involve weight updates to corroborate our theoretical analysis. These methods include *Magnitude* pruning (Zhu & Gupta, 2017) and *Wanda* (Sun et al., 2024). Furthermore, we also do an ablation on methods with weight updates (*SparseGPT* (Frantar & Alistarh, 2023)) for further insights.

**Pruning.** The pruning strategy follows a layer-wise approach, which can be easily augmented with more complex procedures that assign different weights to each layer (Xu et al., 2024; Agarwal et al., 2024). The main focus is on unstructured pruning, where any weight in a matrix may be pruned. Additionally, the structured N:M pruning scenario will also be evaluated. In N:M structure pruning, N weights must be pruned out of every M weights (Hubara et al., 2021b). In particular, the 2:4 and 4:8 structured pruning schemes proposed by Nvidia (Mishra et al., 2021) for faster inference are adopted.

### 4.1 LARGE LANGUAGE MODELING PRUNING

Table 2 reports the perplexity of Llama and Qwen models with various pruning methods. Notice that *STADE* outperforms the other methods consistently across the different pruning scenarios. These results follow our formal analysis, validating our theoretical understanding. Notice that the LLMs in Table 2 use RMSNorm (Zhang & Sennrich, 2019) and therefore we do not use *STADE-W*. Since no layer receives a normalized input, it is no different from standard *STADE*.

Table 2: Perplexity on Wikitext2 for different Llama and Qwen models. C4 dataset is used as the calibration dataset during the pruning process. 2:4 and 4:8 sparsity refers to a structure pruning approach where 2/4 weights are pruned out of every 4/8 weights (Mishra et al., 2021)

| Methods | Sparsity | Llama-1 | Llama-2 | | Llama-3 | | Qwen3 | | |
| | | 7B | 7B | 13B | 3.0-8B | 3.1-8B | 8B | 14B | 32B |
|---|---|---|---|---|---|---|---|---|---|
| Dense | 0% | 5.68 | 5.47 | 4.88 | 6.14 | 6.24 | 9.72 | 8.64 | 7.6 |
| Magnitude | 2:4 | 42.53 | 37.76 | 8.89 | 2401.18 | 792.83 | 294.48 | 38.58 | 29.89 |
| Wanda | 2:4 | 11.52 | 12.12 | 8.98 | 24.31 | 22.87 | 16.41 | 13.14 | 10.33 |
| STADE | 2:4 | **11.38** | **10.82** | **8.42** | **22.30** | **20.52** | **15.20** | **12.52** | **10.13** |
| Magnitude | 4:8 | 16.83 | 15.91 | 7.32 | 181.47 | 212.46 | 115.48 | 21.18 | 36.49 |
| Wanda | 4:8 | **8.57** | 8.60 | 7.00 | 14.61 | 13.78 | 13.24 | 11.12 | 9.46 |
| STADE | 4:8 | 8.63 | **8.29** | **6.86** | **13.69** | **12.93** | **12.62** | **10.69** | **9.29** |
| Magnitude | 50% | 17.29 | 16.03 | 6.83 | 205.45 | 134.28 | 54.56 | 15.22 | 49.09 |
| Wanda | 50% | **7.26** | **6.92** | 5.97 | 9.83 | 9.65 | 11.35 | 10.00 | **8.63** |
| STADE | 50% | 7.43 | 6.97 | **5.95** | **9.63** | **9.47** | **11.19** | **9.60** | 8.65 |

## 4.2 PRUNING REQUIREMENTS OF DIFFERENT LAYERS

In order to see the effect of the normalization layers, we investigate OPT models which use *Layernorm* (Ba, 2016) instead of *RMSNorm* (Zhang & Sennrich, 2019). Since the linear layers that receive the input after a *Layernorm* would be centered, a small ablation is done on the difference when pruning a layer with a centered input vs an uncentered input (Fig. 1).

Figure 1: When pruning only a specified layer type, different pruning methods can performed better depending on the layer selected, i.e., the pruning metric and the layer pruned are not independent. In particular, we highlight the importance of the input's statistics (such as the mean) when selecting the pruning method.

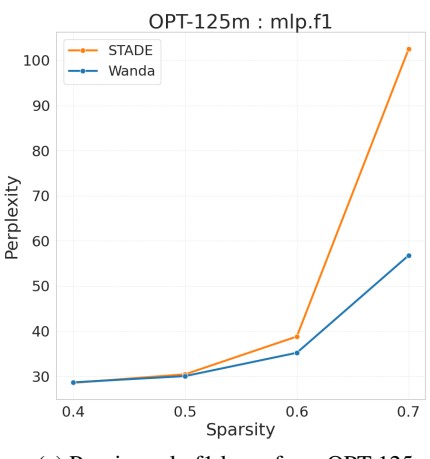

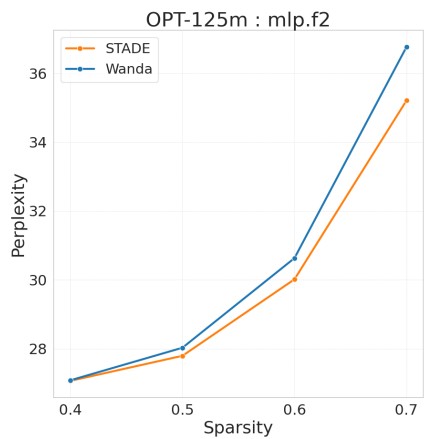

(a) Pruning mlp.f1 layer from OPT-125m (centered input)

(b) Pruning mlp.f2 layer from OPT-125m (uncentered input)

The experiment show that different layers benefit from different pruning methods. In particular, when a layer receives a centered input (first layer of the MLP block), *Wanda* performs better since it already assumes this scenario while *STADE* approximates the mean with the inputs. However, whenever the input is not centered *Wanda* is not able to keep up with *STADE* (second layer of the MLP block). This result is in line with our theoretical analysis and validates our characterization of the pruning problem. With these finding we propose *STADE-W*, a method that combines both

*STADE* and *Wanda*. It uses *Wanda* when the input is centered and *STADE* otherwise. The results in Table 3 show that *STADE-W* improves model performance over *STADE* or *Wanda* when used individually on the OPT family.

Table 3: Perplexity on Wikitext2 with C4 as the calibration dataset.

| Methods | Sparsity | OPT | | | | | | |
|---------|----------|------|------|------|------|------|------|------|
| | | 125m | 350m | 1.3b | 2.7b | 6.7b | 13b | 30b |
| Dense | 0% | 27.65 | 22.00 | 14.62 | 12.47 | 10.86 | 10.13 | 9.56 |
| Magnitude | 2:4 | 341.46 | 417.01 | 427.09 | 1152.92 | 264.04 | 484.64 | 1981.10 |
| Wanda | 2:4 | 80.24 | 113.54 | 28.23 | 21.20 | 15.89 | **15.52** | 13.44 |
| STADE | 2:4 | 109.68 | 100.16 | **27.19** | 24.08 | 16.44 | 17.61 | 15.35 |
| STADE-W | 2:4 | **76.08** | **99.82** | 27.55 | **20.68** | **15.64** | 15.57 | **12.40** |
| Magnitude | 4:8 | 169.09 | 160.73 | 240.13 | 166.93 | 196.15 | 450.06 | 564.03 |
| Wanda | 4:8 | 53.18 | 58.49 | 22.15 | 16.77 | 13.56 | 13.37 | 10.88 |
| STADE | 4:8 | 68.19 | 57.62 | **21.34** | 17.38 | 13.79 | 14.98 | 11.42 |
| STADE-W | 4:8 | **52.64** | **56.69** | 21.93 | **16.66** | **13.41** | **13.34** | **10.85** |
| Magnitude | 50% | 193.35 | 97.78 | 1713.49 | 265.17 | 968.77 | 11609.08 | 168.09 |
| Wanda | 50% | **38.94** | 36.21 | 18.42 | 14.22 | 11.98 | **11.92** | **10.03** |
| STADE | 50% | 49.04 | 37.51 | **17.75** | 14.36 | **11.87** | 13.10 | 10.19 |
| STADE-W | 50% | 39.22 | **36.15** | 18.38 | **14.20** | 11.97 | 11.96 | 10.05 |

## 4.3 ZERO-SHOT COMPARISON

While model perplexity serves as an important evaluation of pruning strategies, measuring prediction accuracy is equally crucial for large language models and their pruned variants. To test the impact of the different pruning methods on model accuracy, we evaluate on multiple zero-shot tasks across different datasets. The summarized results are reported in Table 4.

Table 4: Zero shot accuracy averaged over 8 individual tasks (*Arc-Challenge*, *Arc-Easy*, *Boolq*, *HellaSwag*, *OpenBookQA*, *RTE-3*, *WinoGrande* and *MMLU*. The results for each individual tasks can be found in the appendix.

| Method | Sparsity | Qwen3-0.6B | Qwen3-1.7B | Qwen3-4B | Qwen3-8B | Qwen3-14B |
|--------|----------|------------|------------|----------|----------|-----------|
| Dense | 0% | 45.73% | 56.42% | 63.52% | 66.63% | 69.48% |
| Magnitude | 50% | 32.91% | 33.63% | 35.40% | 41.11% | 60.65% |
| Wanda | 50% | 40.25% | 49.85% | **57.66%** | 62.48% | 67.23% |
| STADE | 50% | **40.36%** | **50.38%** | 57.10% | **62.99%** | **67.64%** |
| Magnitude | 2:4 | 30.21% | 33.09% | 33.11% | 33.62% | 48.54% |
| Wanda | 2:4 | 33.28% | 38.57% | **47.28%** | 54.71% | 59.81% |
| STADE | 2:4 | **33.85%** | **39.72%** | 43.79% | **56.62%** | **59.94%** |
| Magnitude | 4:8 | 31.17% | 33.75% | 35.43% | 34.26% | 54.53% |
| Wanda | 4:8 | **35.49%** | 43.21% | **53.85%** | 60.17% | 63.33% |
| STADE | 4:8 | 34.89% | **43.26%** | 47.99% | **60.52%** | **63.69%** |

The results on the zero-shot task align with those observed when evaluating perplexity (Table 2). *STADE* method demonstrates competitive performance across a range of models and tasks.

## 4.4 WEIGHT UPDATE IMPORTANCE

Pruning methods can also be improved by updating the unpruned weights, as done by *SparseGPT*. Nevertheless, this makes the pruning process slower and more computationally demanding (Sec. A.1). In this section, we compare the performance of *SparseGPT* against *STADE*, with a particular focus on the critical importance of its weight update mechanism.

Table 5: Perplexity comparison with pruning methods that update weights (*SparseGPT*).

| Model | Sparsity | Magnitude | Wanda | SparseGPT | SparseGPT (w/o upd.) | STADE |
|---|---|---|---|---|---|---|
| Qwen3-1.7B | 0% | | | 16.67 | | |
| Qwen3-1.7B | 2:4 | 1808.24 | 61.63 | **31.74** | 51.75 | 46.90 |
| Qwen3-1.7B | 4:8 | 614.71 | 32.62 | **25.38** | 28.80 | 27.76 |
| Qwen3-1.7B | 50% | 174.10 | 20.63 | 23.73 | 22.89 | **18.67** |
| Qwen3-8B | 0% | | | 9.72 | | |
| Qwen3-8B | 2:4 | 294.48 | 16.41 | **14.48** | 14.99 | 15.20 |
| Qwen3-8B | 4:8 | 115.48 | 13.24 | 12.65 | 13.02 | **12.62** |
| Qwen3-8B | 50% | 54.56 | 11.35 | 11.49 | 11.80 | **11.19** |

Table 5 shows that *SparseGPT* and *STADE* are the best performing method. Nevertheless, when *SparseGPT* is not allowed to update the unpruned weights (*SparseGPT (w/o upd.)*), the performance drops below of *STADE*, showing that *STADE* has better performance than any other method when the unpruned weights are frozen, following the theoretical results (Sec. 3).

## 5 CONCLUSION

This work presents a comprehensive analysis of optimal weight pruning in neural networks and provides a theoretical framework that explains why *Wanda* is effective in many common deep learning scenarios. It demonstrates that while *Wanda* performs optimally in layers with centered inputs, its effectiveness diminishes in layers that receive uncentered inputs. In response to these observations, we propose a new pruning criterion (*STADE*) that handles this scenario. We demonstrate theoretically and empirically that *STADE* outperforms *Wanda* for uncentered inputs.

We also observe that different layers have different input statistics and therefore the optimal pruning criterion might change between layers. Building upon these insights, we introduce *STADE-W*, which dynamically combines *Wanda* and *STADE* based on the input statistics of each layer, making it, to the best of current knowledge, the first pruning method that employs different pruning criterions for different layers. We do extensive experiments on Qwen, Llama and Open Pre-trained Transformers models. We not only evaluate perplexity but also zero-shot performance. The results validate our theoretical analysis and reveal that pruning effectiveness varies according to the input characteristics of each layer. In particular, we show how layers with normalized inputs are better pruned with *Wanda* criterion while wuth uncentered inputs *STADE* outperforms *Wanda*. Moreover, our experiments demonstrate that incorporating weight update mechanisms (as exemplified by *SparseGPT*) can improve performance, further highlighting the benefits of updating the unpruned weights and a future research direction.

All together, these contributions not only advance the understanding of pruning strategies but also offer a new robust method for reducing the computational demands of large language models without significant performance loss. The insights provided herein pave the way for more efficient deployment of large-scale models in resource-constrained environments.

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

## A APPENDIX

### A.1 PRUNING TIME

Different pruning methods take different time to calculate their corresponding pruning scores. We report in Table 6 the pruning time for the different methods in different pruning scenarios.

Table 6: Pruning time comparison in seconds for different methods on Llama-3.2-1B.

| Sparsity | Wanda | STADE | STADE-W | SparseGPT |
|----------|-------|-------|---------|-----------|
| 50%      | 72,89 | 72,02 | 74,55   | 222,31    |
| 2:4      | 77,87 | 74,80 | 73,04   | 204,52    |
| 4:8      | 70,10 | 73,39 | 71,51   | 215,36    |

Notice that Table 6 shows how *SparseGPT* triples the pruning time. This is due to the unpruned weight update which requires inverse matrix calculation which would become more expensive as the layers becomes wider.

## A.2 Implementation details

When estimating the mean and the standard deviation, loading the full data requires a huge amount of memory resources. Therefore, the mean and standard deviation is calculated in a dynamic manner with 1 mini-batch of data at a time.

Calculating the sum of squares results in high values which lead to values that are no longer updated for updates that fall outside of the mantisa range for floating point numbers. In order to avoid that, the mean and standard deviation is calculated in each iteration as follows:

---

**Algorithm 1** Mean and variance calculation

---

**Input**: dataloader $D$
**Output**: mean $\mu$, var $\mathbb{V}$

1: Let $N = 0$, $\mu = 0$ and $\mathbb{V} = 0$.
2: **for** $x_{batch}$ in $D$ **do**
3:     $N_{new} = N + len(x_{batch})$
4:     $\mu_{new} = \frac{\mu*N}{N_{new}} + \frac{sum(x\_batch)}{N_{new}}$
5:     $\mathbb{V} = \frac{(N-1)*\mathbb{V}}{N_{new}-1} + \frac{N*\mu^2}{(N_{new}-1)} - \frac{N_{new}*\mu_{new}^2}{(N_{new}-1)} + \frac{sum(x_{batch}^2)}{N_{new}-1}$
6:     $N = N_{new}$
7:     $\mu = \mu_{new}$
8: **end for**
9: **return** $\mu$, $\mathbb{V}$

---

## A.3 Training details

While the pruning methods shown don't have hyperparameter to tuned, there are some training details that we would like to mention:

- **C4 calibration dataset**: Following *Wanda*, when using C4 dataset only the file *'en/c4-train.00000-of-01024.json.gz'* is used during pruning to speed up the process. the full dataset can be fined in *'https://huggingface.co/datasets/allenai/c4/tree/main/en'*.

- **Sequence length**: Some LLMs allow over 10k context window. In order to run the models in hardware constrains scenarios, the sequence length is cropped to 2048 experiment on more models. This is both applied during pruning and evaluation.

## A.4 Intuitive explanation of STADE

In this section a simplified explanation for *STADE* will be shown. We will assume that the input multivariate distribution $X \in \mathbb{R}^2$ is normally distributed, i.e., $X_i \sim \mathcal{N}(\mu_i, \sigma_i)$. **Notice that STADE does not require the input to be normally distributed,** this is just a simplification for the sake of the explanation. In the same way as in the Methodology section, we will consider the pruning problem for one column. In this case the corresponding output of the linear layer ($\hat{y}$) can be calculated as:

$$\begin{aligned}
\hat{y} &= B + x_1 W_1 + x_2 W_2 \\
&= B + (\mu_1 + \epsilon_1 \sigma_1) W_1 + (\mu_2 + \epsilon_2 \sigma_2) W_2 \\
&= (B + \mu_1 + \mu_2) + \sigma_1 W_1 \epsilon_1 + \sigma_2 W_2 \epsilon_2
\end{aligned} \tag{10}$$

Notice that $\epsilon_1, \epsilon_2 \sim \mathcal{N}(0, 1)$ and therefore will affect the same way when pruning the weight. However, when deciding whether to prune $W_1$ or $W_2$, it's not only about the value of the weight but also about the standard deviation of the corresponding input. This is due to the fact that the mean of the input can be added to the bias and therefore omitted when pruning the weights.

## A.5 BIAS USAGE ABLATION

*STADE* method updates the bias term when pruning the models. In models like OPT which already have bias, this a is reasonable assumption. However, Llama and Qwen models do not have a bias term and therefore. Applying *STADE* on those models would result in adding a new bias term which could be considered as adding an extra weight variable. This could be considered an unfair advantage when compared to the other methods. To investigate this, a small ablation is done where the bias term is not update.

Table 7: Ablation on the importance of the bias update in *STADE* algorithm.

| Method | STADE | STADE (w/o bias) | STADE | STADE (w/o bias) | STADE | STADE (w/o bias) |
|---|---|---|---|---|---|---|
| Sparsity | 0.5 | | 2:4 | | 4:8 | |
| Llama-7B | **7.43** | **7.43** | **11.38** | **11.38** | **8.63** | **8.63** |
| Llama2-7B | **6.97** | **6.97** | **10.82** | **10.82** | **8.29** | **8.29** |
| Llama2-13B | **5.95** | **5.95** | **8.42** | **8.42** | **6.86** | **6.86** |
| Llama3-8B | **9.63** | **9.63** | **22.30** | 22.37 | **13.69** | 14.55 |
| Llama3.1-8B | **9.47** | **9.47** | **20.52** | **20.52** | 12.93 | **12.91** |

The results shown in Tab. 7 and 8 demonstrate that there is little to no difference when adding the bias term and if one wants to remove this term the results are almost identical. Empirically we observe that for any layer, the sum of the absolute terms of the bias is always smaller than $10^{-2}$, which explains why removing it has little to no impact.

## A.6 STADE* VARIATION DERIVATION

Following the results from Sec. A.5 and taking into acount the fact that not all models use a bias term in their linear layers, a variation of *STADE* can be formulated without the possibility of updating the bias (*STADE\**), i.e., the linear layer to prune has no bias term and the pruning method is not allowed to add a bias term in order to keep the model structure. To do so we expand on the previous derivations from the main paper as follows:

$$
\begin{aligned}
\min_{W^*,0} \mathbb{E}\Big[ \Big( (B + \sum_{i=1}^{M} X_i W_i) - (B^* + \sum_{i=1}^{M} X_i W_i^*) \Big)^2 \Big] \\
= \min_{j,0} (B - B^*)^2 + 2(B - B^*)\mu_j W_j + (\sigma_j^2 + \mu_j^2)W_j^2 \\
= \min_{j} (\sigma_j^2 + \mu_j^2)W_j^2 \\
\approx \Big[ ||X_{:,j} - \frac{1}{N}\sum_{n=1}^{N} X_{n,j}||_2^2 + (\frac{1}{N}\sum_{n=1}^{N} X_{n,j})^2 \Big] |W_{i,j}|^2
\end{aligned}
\tag{11}
$$

Standard *STADE* had better performance than *STADE\** even when not updating the bias, i.e., the pruning criterion even though locally optimal (optimal for the linear layer pruning) is not optimal globally (for the model pruning). This can be observed in tables 8 to 16. The only cases where *STADE\** outperforms *STADE* is when *STADE* has a big spike/jump on the perplexity. It seems that it is performing worse in general, but it has always consistent results avoiding the huge spikes. Future research should investigate this phenomena to further improve the pruning methodology.

## A.7 LLM USAGE FOR PAPER WRITING

In the writing of this paper LLMs were used to polish existing text to be more cohesive and coherent.

## A.8 ADDITIONAL EXPERIMENTS

Tables 8 to 16 show experiments on more models and additional pruning metrics measuring both perplexity and zero-shot performance.

Table 8: Wikitext perplexity for the Qwen family. Notice that *STADE-W* here is applied after the *RMSnorm* layers. As explained in the Methodology, Qwen3 uses *RMSnorm* which does not normalize the inputs and therefore it is not applicable as it was with the OPT family. We observe huge spikes for Qwen3-0.6B and Qwen3-4B. To the best of our knowledge the only difference with the other models is the usage of tie-embeddings. Nevertheless, Qwen3-1.7B also uses them and doesn't exhibit those spikes. We were not able to identify the source behind these spikes.

| Methods | Sparsity | Qwen3 | | | | |
|---|---|---|---|---|---|---|
| | | 0.6B | 1.7B | 4B | 8B | 14B |
| Dense | 0% | 20.95 | 16.67 | 13.64 | 9.72 | 8.64 |
| Magnitude | 2:4 | 85481.66 | 1808.24 | 1970.45 | 294.48 | 38.58 |
| Wanda | 2:4 | 190.03 | 61.63 | 30.17 | 16.41 | 13.14 |
| STADE | 2:4 | 13785.28 | 46.90 | 133.17 | 15.20 | 12.52 |
| STADE (w/o bias) | 2:4 | 13785.28 | 46.90 | 131.50 | 15.20 | 12.51 |
| STADE* | 2:4 | 193.29 | 60.66 | 30.25 | 16.41 | 13.10 |
| STADE-W | 2:4 | 171.60 | 47.24 | 32.37 | 15.54 | 12.57 |
| SparseGPT | 2:4 | **91.05** | **31.74** | **21.32** | **14.48** | 12.47 |
| SparseGPT (no update) | 2:4 | 6278.69 | 51.75 | 86.57 | 14.99 | **12.18** |
| Magnitude | 4:8 | 99815.32 | 614.71 | 150.43 | 115.48 | 21.18 |
| Wanda | 4:8 | 73.71 | 32.62 | 22.25 | 13.24 | 11.12 |
| STADE | 4:8 | 304.30 | 27.76 | 51.13 | **12.62** | 10.69 |
| STADE (w/o bias) | 4:8 | 304.30 | 27.79 | 52.95 | 12.63 | 10.68 |
| STADE* | 4:8 | 74.73 | 32.46 | 22.50 | 13.24 | 11.12 |
| STADE-W | 4:8 | 77.89 | 28.67 | 22.57 | 12.71 | **10.67** |
| SparseGPT | 4:8 | **60.55** | **25.38** | **18.64** | 12.65 | 11.16 |
| SparseGPT (no update) | 4:8 | 513.65 | 28.80 | 51.60 | 13.02 | 10.71 |
| Magnitude | 50% | 1455.57 | 174.10 | 111.22 | 54.56 | 15.22 |
| Wanda | 50% | 34.20 | 20.63 | 16.39 | 11.35 | 10.00 |
| STADE | 50% | 34.01 | 18.67 | 16.90 | 11.19 | 9.60 |
| STADE (w/o bias) | 50% | 34.04 | **18.66** | 16.90 | 11.19 | 9.60 |
| STADE* | 50% | 34.06 | 20.57 | 16.39 | 11.35 | 10.01 |
| STADE-W | 50% | **33.96** | 18.98 | **16.04** | **11.10** | **9.54** |
| SparseGPT | 50% | 34.14 | 23.73 | 17.39 | 11.49 | 10.08 |
| SparseGPT (no update) | 50% | 89.12 | 22.89 | 20.03 | 11.80 | 9.95 |

Table 9: Zero-shot performance on Arc Challenge (Clark et al., 2018).

| Method | Sparsity | Qwen3-0.6B | Qwen3-1.7B | Qwen3-4B | Qwen3-8B | Qwen3-14B |
|--------|----------|------------|------------|----------|----------|-----------|
| Dense | 0% | 31.40% | 39.76% | 50.77% | 55.80% | 58.62% |
| Magnitude | 50% | 20.90% | 19.11% | 22.70% | 28.07% | 48.81% |
| Wanda | 50% | 23.38% | 30.46% | 39.76% | 50.85% | **55.20%** |
| STADE | 50% | **23.98%** | **33.53%** | **41.72%** | **51.88%** | 55.03% |
| Magnitude | 2:4 | 20.65% | 20.56% | 22.44% | 18.69% | 37.12% |
| Wanda | 2:4 | 19.71% | 19.71% | **32.17%** | 38.65% | 42.24% |
| STADE | 2:4 | **21.67%** | **20.90%** | 29.86% | **42.32%** | **44.54%** |
| Magnitude | 4:8 | 20.56% | 21.50% | 23.46% | 19.45% | 44.37% |
| Wanda | 4:8 | 19.71% | 24.91% | **38.05%** | 46.16% | 50.60% |
| STADE | 4:8 | **21.25%** | **26.37%** | 33.11% | **47.95%** | **51.96%** |

Table 10: Zero-shot performance on Arc Easy (Clark et al., 2018).

| Method | Sparsity | Qwen3-0.6B | Qwen3-1.7B | Qwen3-4B | Qwen3-8B | Qwen3-14B |
|--------|----------|------------|------------|----------|----------|-----------|
| Dense | 0% | 60.90% | 72.22% | 80.51% | 83.46% | 84.22% |
| Magnitude | 50% | 28.75% | 34.01% | 47.94% | 58.12% | 76.47% |
| Wanda | 50% | 48.65% | 62.12% | **72.90%** | 80.09% | 81.31% |
| STADE | 50% | **48.70%** | **64.31%** | 72.39% | **80.43%** | **81.94%** |
| Magnitude | 2:4 | 26.52% | 28.91% | 33.46% | 36.57% | 63.09% |
| Wanda | 2:4 | **32.79%** | 47.35% | **59.26%** | 72.69% | **72.47%** |
| STADE | 2:4 | 27.86% | **49.71%** | 50.17% | **74.03%** | 71.42% |
| Magnitude | 4:8 | 27.48% | 30.51% | 42.63% | 40.32% | 72.35% |
| Wanda | 4:8 | **40.74%** | 56.65% | **67.89%** | 77.23% | 78.41% |
| STADE | 4:8 | 31.65% | **58.25%** | 57.58% | **77.78%** | **78.87%** |

Table 11: Zero-shot performance on Boolq (Clark et al., 2019).

| Method | Sparsity | Qwen3-0.6B | Qwen3-1.7B | Qwen3-4B | Qwen3-8B | Qwen3-14B |
|--------|----------|------------|------------|----------|----------|-----------|
| Dense | 0% | 64.53% | 77.46% | 85.11% | 86.64% | 89.33% |
| Magnitude | 50% | 46.36% | 46.94% | 38.32% | 52.32% | 79.60% |
| Wanda | 50% | **62.20%** | **73.61%** | **82.51%** | **84.86%** | 88.07% |
| STADE | 50% | 62.08% | 73.43% | 80.52% | 84.68% | **88.20%** |
| Magnitude | 2:4 | 38.65% | 50.28% | 46.94% | 43.33% | 65.78% |
| Wanda | 2:4 | 46.76% | 63.85% | **73.39%** | **82.17%** | 85.81% |
| STADE | 2:4 | **52.75%** | **67.22%** | 70.18% | 81.68% | **86.64%** |
| Magnitude | 4:8 | 42.51% | 51.10% | 42.08% | 41.28% | 68.59% |
| Wanda | 4:8 | 48.44% | **70.83%** | **78.41%** | **85.11%** | **87.43%** |
| STADE | 4:8 | **57.68%** | 66.33% | 74.50% | 84.92% | **87.43%** |

Table 12: Zero-shot performance on HellaSwag (Zellers et al., 2019).

| Method | Sparsity | Qwen3-0.6B | Qwen3-1.7B | Qwen3-4B | Qwen3-8B | Qwen3-14B |
|---|---|---|---|---|---|---|
| Dense | 0% | 37.55% | 46.12% | 52.27% | 57.14% | 60.97% |
| Magnitude | 50% | 26.07% | 28.16% | 29.79% | 34.51% | 49.76% |
| Wanda | 50% | 32.24% | 38.56% | 45.03% | 50.08% | 55.11% |
| STADE | 50% | **32.86%** | **40.26%** | **45.92%** | **51.65%** | **56.66%** |
| Magnitude | 2:4 | 25.70% | 26.56% | 27.03% | 26.98% | 42.16% |
| Wanda | 2:4 | **27.17%** | 29.90% | **37.46%** | 42.35% | 48.00% |
| STADE | 2:4 | 26.51% | **31.08%** | 36.22% | **43.76%** | **49.53%** |
| Magnitude | 4:8 | 26.21% | 26.89% | 30.43% | 28.04% | 45.68% |
| Wanda | 4:8 | **29.09%** | 33.91% | **41.22%** | 46.26% | 51.69% |
| STADE | 4:8 | 28.27% | **35.16%** | 40.10% | **47.91%** | **53.08%** |

Table 13: Zero-shot performance on OpenbookQA (Mihaylov et al., 2018).

| Method | Sparsity | Qwen3-0.6B | Qwen3-1.7B | Qwen3-4B | Qwen3-8B | Qwen3-14B |
|---|---|---|---|---|---|---|
| Dense | 0% | 21.00% | 28.40% | 29.20% | 31.00% | 35.00% |
| Magnitude | 50% | 15.00% | 13.40% | 13.80% | 18.60% | 30.40% |
| Wanda | 50% | 16.60% | 21.20% | 26.20% | 28.60% | 33.00% |
| STADE | 50% | **18.00%** | **22.80%** | **26.80%** | **30.60%** | **33.60%** |
| Magnitude | 2:4 | 14.00% | 13.80% | 13.00% | 14.80% | 26.20% |
| Wanda | 2:4 | 13.00% | 13.60% | 22.20% | 23.00% | 28.60% |
| STADE | 2:4 | **15.40%** | **15.40%** | **22.80%** | **23.20%** | **30.20%** |
| Magnitude | 4:8 | 13.20% | 14.40% | 14.20% | 15.40% | 28.20% |
| Wanda | 4:8 | 15.60% | 17.40% | **24.80%** | 26.40% | 31.40% |
| STADE | 4:8 | **15.80%** | **17.80%** | 22.80% | **28.40%** | **31.60%** |

Table 14: Zero-shot performance on RTE (Bentivogli et al., 2009).

| Method | Sparsity | Qwen3-0.6B | Qwen3-1.7B | Qwen3-4B | Qwen3-8B | Qwen3-14B |
|---|---|---|---|---|---|---|
| Dense | 0% | 54.15% | 70.76% | 75.81% | 78.34% | 77.62% |
| Magnitude | 50% | 51.26% | 52.71% | 51.62% | 52.71% | 69.31% |
| Wanda | 50% | **54.15%** | **70.04%** | **72.92%** | 70.04% | 81.23% |
| STADE | 50% | 51.26% | 67.15% | 67.51% | **70.76%** | **82.31%** |
| Magnitude | 2:4 | 42.96% | 49.46% | 47.29% | 52.71% | 48.74% |
| Wanda | 2:4 | 51.62% | 52.35% | **55.96%** | 61.73% | **70.40%** |
| STADE | 2:4 | **53.07%** | **52.71%** | 54.87% | **69.31%** | 65.70% |
| Magnitude | 4:8 | 46.21% | 51.26% | 52.71% | 52.35% | 54.87% |
| Wanda | 4:8 | **53.07%** | 52.35% | **71.12%** | **72.92%** | **70.04%** |
| STADE | 4:8 | 47.65% | **53.07%** | 55.23% | 70.76% | 69.68% |

Table 15: Zero-shot performance on Winogrande (Keisuke et al., 2019).

| Method | Sparsity | Qwen3-0.6B | Qwen3-1.7B | Qwen3-4B | Qwen3-8B | Qwen3-14B |
|---|---|---|---|---|---|---|
| Dense | 0% | 56.20% | 60.93% | 66.06% | 67.72% | 72.85% |
| Magnitude | 50% | 49.88% | 51.62% | 51.62% | 55.09% | 64.80% |
| Wanda | 50% | 54.06% | **57.38%** | 62.12% | **69.61%** | **72.77%** |
| STADE | 50% | **55.41%** | 56.91% | **62.19%** | 68.35% | 71.82% |
| Magnitude | 2:4 | 48.78% | 52.09% | 51.30% | 51.78% | 61.01% |
| Wanda | 2:4 | **52.25%** | **53.67%** | **56.98%** | 62.43% | 68.27% |
| STADE | 2:4 | 50.12% | 52.33% | 53.75% | **63.85%** | **68.75%** |
| Magnitude | 4:8 | 49.80% | 50.20% | 51.38% | 51.93% | 64.33% |
| Wanda | 4:8 | **52.80%** | **53.99%** | **61.01%** | **66.61%** | **69.85%** |
| STADE | 4:8 | 52.09% | 53.75% | 56.98% | 65.43% | 69.22% |

Table 16: Zero-shot performance on MMLU (Hendrycks et al., 2021).

| Method | Sparsity | Qwen3-0.6B | Qwen3-1.7B | Qwen3-4B | Qwen3-8B | Qwen3-14B |
|---|---|---|---|---|---|---|
| Dense | 0% | 56.20% | 60.93% | 66.06% | 67.72% | 72.85% |
| Magnitude | 50% | 49.88% | 51.62% | 51.62% | 55.09% | 64.80% |
| Wanda | 50% | 54.06% | **57.38%** | 62.12% | **69.61%** | **72.77%** |
| STADE | 50% | **55.41%** | 56.91% | **62.19%** | 68.35% | 71.82% |
| Magnitude | 2:4 | 48.78% | 52.09% | 51.30% | 51.78% | 61.01% |
| Wanda | 2:4 | **52.25%** | **53.67%** | **56.98%** | 62.43% | 68.27% |
| STADE | 2:4 | 50.12% | 52.33% | 53.75% | **63.85%** | **68.75%** |
| Magnitude | 4:8 | 49.80% | 50.20% | 51.38% | 51.93% | 64.33% |
| Wanda | 4:8 | **52.80%** | **53.99%** | **61.01%** | **66.61%** | **69.85%** |
| STADE | 4:8 | 52.09% | 53.75% | 56.98% | 65.43% | 69.22% |

