# OpenReview forum: "STADE: Standard Deviation as a Pruning Metric"
_ICLR.cc/2026/Conference — ICLR 2026 Conference Withdrawn Submission_

### Official Review · Reviewer_xyj3 · 2025-10-31

**Soundness:** 2
**Presentation:** 2
**Contribution:** 2
**Rating:** 2
**Confidence:** 3

**Summary:**

This paper proposes **STADE** (and **STADE-W**) as pruning criteria derived from a formalization of the *layer output reconstruction* objective. Starting from a one-column pruning problem that minimizes the expected squared difference between pre- and post-pruning layer outputs—allowing only the bias to change—the authors derive that the optimal feature to prune is the one minimizing \(\sigma_j^2 W_j^2\). This yields a criterion proportional to the **standard deviation** of the layer input times the weight magnitude; when inputs are centered (\(\mu=0\)), the derivation collapses to **Wanda**’s \( \lVert X_{:,j}\rVert_2 |W_{j}| \) rule, thereby explaining conditions where Wanda is optimal and where it is not. The paper then proposes **STADE-W**, which **switches** between STADE (uncentered inputs) and Wanda (centered inputs) per layer. Experiments on LLaMA/Qwen/OPT report perplexity and zero-shot results for unstructured and N:M (2:4, 4:8) sparsity; OPT ablations illustrate that different layers (e.g., MLP first vs. second linear) prefer different criteria.

**Strengths:**

- **Clean theory → criterion mapping.** The paper connects a simple reconstruction objective to practical pruning rules, explaining **when** Wanda is optimal (centered inputs) and **why** a mean-aware criterion is needed otherwise.
- **Layer-aware insight.** The OPT ablation shows **different layers** (e.g., MLP f1 vs. f2) benefit from different metrics, motivating **STADE-W** and giving practitioners actionable guidance.
- **Broad model coverage & metrics.** Results span LLaMA/Qwen/OPT with both **unstructured** and **N:M** sparsity, reporting perplexity and **LM-Harness** zero-shot accuracy.
- **Pruning-time reporting.** The appendix shows pruning-time overhead is close to Wanda and **far below** SparseGPT’s update-based routine, which is useful for adoption.

**Weaknesses:**

1. **Minimal novelty over Wanda; activation-outlier story is underdeveloped.** The method is largely incremental over Wanda, switching from \(\lVert X\rVert_2\) to a de-meaned statistic and adding a layer-dependent gate. Wanda’s effectiveness in LLMs is often credited to activation outliers; please analyze how STADE/STADE-W interacts with outlier channels (e.g., tail-mass concentration, per-channel kurtosis) and whether your criterion preserves the same “keep-the-outliers” effect that underpins Wanda’s success. Offer concrete insights beyond the algebraic substitution.
2. **Sparsity ≠ efficiency; missing speedups.** The paper reports sparsity/perplexity/accuracy but no end-to-end speedups under realistic sparse runtimes. Unstructured sparsity (and even N:M without vendor kernels) rarely yields proportional acceleration.
3. **Theory hinges on output-matching with bias update.** The optimality proof assumes that keeping the layer output nearly unchanged (while letting the bias absorb the mean term) correlates with global optimality. In LLM pruning, prior attempts optimizing this surrogate have mixed results. Please justify this assumption empirically (e.g., correlation between the objective and downstream metrics across layers/sparsities) and clarify the role of bias updates; your own ablations suggest small impact in some cases, which weakens the premise.
4. **Practical region underexplored.** Many tables emphasize moderate to high sparsity, but what matters most is the **0–50%** regime where accuracy stays near-dense. A clear Pareto (accuracy vs. sparsity) in that regime, against Wanda and SparseGPT (no-update), would better establish value.
5. **Transfer of insights to other compressors is unclear.** Beyond Wanda-style post-training pruning, can your statistics improve movement pruning, magnitude with per-layer allocation, or second-order methods (e.g., better pre-screening or layer routing)? Could you discuss how your analysis generalizes to quantization-aware pruning or hybrid prune-then-quantize pipelines?

**Questions:**

1. **Activation outliers.** How does STADE/STADE-W behave on channels with extreme activation magnitudes/outliers? Can you quantify outlier preservation (e.g., fraction of top-k activation channels retained) vs. Wanda across layers, and relate this to accuracy retention?
2. **Real speedups.** Under a production sparse runtime, can you report latency, tokens/s for dense vs. Wanda vs. STADE/STADE-W across unstructured and 2:4 / 4:8 setups on fixed hardware and decoding parameters?
3. **Bias update & objective validity.** Your theory assumes the bias absorbs the mean term. In architectures without bias in linear layers (e.g., many LLMs), what guarantees remain? Your appendix suggests small differences when forbidding bias updates—does that imply the theoretical objective is not the key driver? Please reconcile.
4. **0–50% sparsity Pareto.** Can you show that STADE/STADE-W strictly dominates Wanda and SparseGPT (no update) in the **practical** sparsity regime, with multi-seed statistics and confidence intervals?

---

### Official Review · Reviewer_oZhv · 2025-10-31

**Soundness:** 2
**Presentation:** 2
**Contribution:** 2
**Rating:** 2
**Confidence:** 4

**Summary:**

This paper proposes STADE, a pruning method derived from an analytical explaination of when the Wanda criterion is optimal. The authors shows that Wanda's L2-based pruning rule is optimal only under zero-mean (centered) input statistics. For non-centered inputs, they derive an adjusted metric proportional to the input standard deviation and proposed STADE-W, which adaptively selects between Wanda and STADE per layer. Experiments on LLaMA, Qwen, and OPT model families show modest improvements in perplexity and zero-shot accuracy over Wanda and magnitude pruning. The authors also include ablations on layer normalization (Fig. 1), bias update effects (Tab. 7), and pruning time (Tab. 6).

**Strengths:**

1. The problem setting and formulation is clear. The derivation from Eq. 2-6 is transparent and interpretable.
2. STADE-W provides a simple, practical rule for distinguishing normalized vs. unnormalized layers.
3. Computational efficiency is good. Table 6 shows pruning time is on par with Wanda and 3× faster than SparseGPT.
4. Evaluations on LLaMA, Qwen, and OPT families demonstrate method applicability across norms.

**Weaknesses:**

1. Weak empirical results. Improvements over Wanda are very small, which is often within statistical noise, not consistently significant (see Table 2, 4). For example, STADE’s perplexity on LLaMA-7B improves only ~0.14 points; in many configurations it underperforms Wanda.

2. Designing pruning metrics is a active field, but this paper fails to compare with recent baselines such as  Pruner-Zero [1] and SymWanda [2]. For example, SymWanda offers a general framework subsuming Wanda and other metrics under unified scaling symmetries.

3. No fine-grained variance analysis. Despite claiming standard deviation–based pruning, the paper does not quantify variance distributions or show correlation plots between σ and pruning efficacy.

4. Insufficient statistical reporting. No standard deviations, error bars, or multiple seeds—making it unclear if improvements are reproducible.

5. Method simplicity vs. claimed novelty. The metric boils down to subtracting mean activations before applying Wanda’s rule, which is conceptually trivial once the mean-bias link is identified.

[1] Pruner-Zero: Evolving Symbolic Pruning Metric from scratch for Large Language Models. ICML 2024.

[2] Symmetric Pruning for Large Language Models. arXiv 2501.

**Questions:**

1. How is “centered input” detected automatically in STADE-W? Is there a numerical threshold for $\lvert \mu_i \rvert$ ?

2. Can STADE be interpreted as a specific instantiation within SymWanda’s theoretical framework? If so, how does it differ analytically?

3. Have you tested whether subtracting the mean during Wanda’s calibration step alone (without a new metric) reproduces the same gains?

---

### Official Review · Reviewer_MwEo · 2025-11-02

**Soundness:** 3
**Presentation:** 2
**Contribution:** 2
**Rating:** 4
**Confidence:** 4

**Summary:**

The paper studies post-training pruning metrics for LLMs. It shows that Wanda’s heuristic, which scores weight $W_{j,i}$ by $\lVert X_{:,j}\rVert_2,|W_{j,i}|$, is theoretically optimal when the incoming activations are centered. From the same quadratic loss setup, the authors derive STADE, which replaces $\lVert X_{:,j}\rVert_2$ with the standard deviation of the input feature, i.e., $\lVert X_{:,j}-\bar X_j\mathbf{1}\rVert_2$, arguing that this is optimal when inputs are not centered. They also propose STADE-W, which picks Wanda or STADE per layer depending on whether inputs are centered. Experiments on LLaMA, Qwen, and OPT report perplexity and zero-shot results, with STADE usually matching or slightly beating Wanda, and STADE-W improving OPT where LayerNorm creates centered inputs. They also compare to SparseGPT and show similar pruning time to Wanda but much faster than SparseGPT.

**Strengths:**

1) Originality: Formalizes a clear condition where Wanda is optimal and derives a simple alternative for uncentered inputs; the per-layer policy (STADE-W) is sensible for architectures mixing normalization behaviors.
2) Clarity: Method is easy to implement: replace $\lVert X\rVert_2$ with the de meaned norm, optionally detect centeredness, keep everything one-shot. Algorithms for streaming mean/variance are provided.
3) Significance: If one insists on no weight updates, a drop-in metric that sometimes improves Wanda at near-zero extra cost is practically useful on commodity pruning pipelines. Results span multiple model families and structured/unstructured sparsities.

**Weaknesses:**

1) Overclaiming “consistency.” Table 2 shows STADE does not dominate Wanda in all settings (e.g., LLaMA-7B 4:8 and 50% shows STADE slightly worse), and improvements are often small. Why SparseGPT is not included in this table?
2) Limited theory scope. The objective ignores cross-feature covariance and multi-column pruning interactions. Extending the analysis to (\sigma)-aware criteria would strengthen the “optimality” story. The appendix’ STADE* variant highlights sensitivity to bias assumptions without a unifying theory.
3) Centeredness detection is hand-wavy. The paper asserts when inputs are centered (LayerNorm) vs not (RMSNorm) but gives no empirical per-layer mean statistics to justify the STADE-W switching policy. Please add layer-wise histograms of means and how the threshold is chosen.
4) No variance estimates or confidence intervals for zero-shot accuracy; several gaps vs Wanda are within likely noise.
5) No end-to-end wall-clock inference benchmarks showing actual speedups/slowdowns on typical hardware.
6) Calibration dependence is underexplored: you use a single C4 shard to compute statistics; sensitivity to sample size or domain shift is not reported.
7) Comparisons. The paper emphasizes “no weight update” regimes, but many practitioners accept light updates; the SparseGPT comparison is fair, yet there’s no look at hybrid first-order updates with STADE-scores, which could be the practical sweet spot.

**Questions:**

1. Layer-wise centeredness: Please report, for each linear layer type, the empirical mean of inputs over the calibration set and the criterion used to decide “centered” vs “uncentered.” How robust is STADE-W to misclassification?
2. Covariance-aware extension: Could you derive an analogue that accounts for cross-feature covariance, e.g., a score proportional to $\sqrt{\mathrm{Var}(X^\top W_j)}$ that uses diagonal-plus-low-rank estimates of $\sigma$?
3. Calibration sensitivity: How do results vary with the number of calibration tokens and with out-of-domain calibration (e.g., WikiText-103 vs C4)? Please include plots of score stability vs sample count.
4. Runtime and speedups: Provide inference throughput and latency on GPUs that support 2:4/4:8 acceleration, to translate perplexity deltas into real deployment wins.
5. Where Wanda wins: In the settings where Wanda outperforms STADE (e.g., some 4:8 cases), what input statistics explain that? Add qualitative diagnostics linking mean/variance to method choice.

---

### Note · Authors · 2025-11-28

I have read and agree with the venue's withdrawal policy on behalf of myself and my co-authors.